# Towards Building More Robust NER Datasets:
# An Empirical Study on NER Dataset Bias from a Dataset Difficulty View

**Ruotian Ma**[1*], **Xiaolei Wang**[1*], **Xin Zhou**[1], **Qi Zhang**[1†], **Xuanjing Huang**[1,2]

[1] School of Computer Science, Fudan University, Shanghai, China
[2] International Human Phenome Institutes, Shanghai, China
{rtma19,qz}@fudan.edu.cn, xlwang22@m.fudan.edu.cn

## Abstract

Recently, many studies have illustrated the robustness problem of Named Entity Recognition (NER) systems: the NER models often rely on superficial entity patterns for predictions, without considering evidence from the context. Consequently, even state-of-the-art NER models generalize poorly to out-of-domain scenarios when out-of-distribution (OOD) entity patterns are introduced. Previous research attributes the robustness problem to the existence of NER dataset bias, where simpler and regular entity patterns induce shortcut learning. In this work, we bring new insights into this problem by comprehensively investigating the NER dataset bias from a dataset difficulty view. We quantify the entity-context difficulty distribution in existing datasets and explain their relationship with model robustness. Based on our findings, we explore three potential ways to de-bias the NER datasets by altering entity-context distribution, and we validate the feasibility with intensive experiments. Finally, we show that the de-biased datasets can transfer to different models and even benefit existing model-based robustness-improving methods, indicating that building more robust datasets is fundamental for building more robust NER systems.

## 1 Introduction

Named Entity Recognition (NER), aiming to recognize named entities from unstructured data, is widely studied by researchers as a fundamental task in Natural Language Processing (NLP) and a crucial task in practical applications (Lample et al., 2016; Chiu and Nichols, 2016; Li et al., 2020). Recently, the advances in pre-trained language models (Devlin et al., 2019; Lewis et al., 2020; Liu et al., 2019) have contributed to promising performance on standard NER benchmarks, such

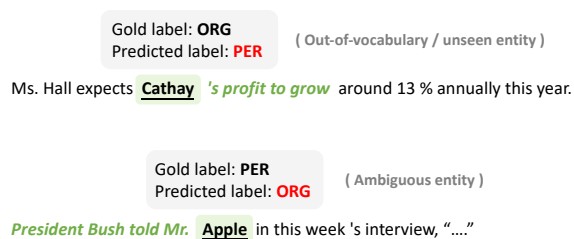

Figure 1: Examples of two typical failures of NER models, copied from the OntoNotes 5.0 dataset.

as CoNLL03 (Tjong Kim Sang and De Meulder, 2003) and OntoNotes 5.0 (Weischedel et al., 2013).

Despite the success, recent research has demonstrated the robustness problem in NER: the state-of-the-art NER systems often rely on superficial entity patterns for predictions, while disregarding evidence from the context. Consequently, the models show poor generalization ability in out-of-domain (OOD) scenarios where out-of-domain entity patterns are introduced (Lin et al., 2020; Ghaddar et al., 2021; Wang et al., 2022). For instance, Figure 1 shows two typical OOD failures of SOTA NER models: (1) Instances with out-of-vocabulary entities (upper). The entity "Cathay" is unseen in the training data, however, the model fails to deduce correctly given obvious evidence "[]'s profit to grow" in the context. (2) Instances with ambiguous entities (bottom). Fu et al. (2020) shows when an entity is labeled differently across domains, the model fails to recognize well even with supportive context, such as mistaking the person name "Apple" in the example.

Previous works have delved into the robustness problem of NER from different perspectives. (Agarwal et al., 2020; Ghaddar et al., 2021; Kim and Kang, 2022; Lin et al., 2020; Wang et al., 2021) examine the NER models by constructing challenging test sets. Their experiments reveal that NER models fail to "learn from context information" during training. Agarwal et al. (2020) and Kim and Kang (2022) evaluate models

---

[*]Equal contribution.
[†]Corresponding authors.

on controlled entity sets and consider the poor robustness is due to the model's tendency to memorize entity patterns. Lin et al. (2020) further designs a series of randomization tests and demonstrates that the strong name regularity and high mention coverage rate in existing NER datasets might hinder model robustness. All these studies indicate that the poor robustness of NER models might be due to a hidden dataset bias of existing NER datasets: the entity pattern is "easier" for the model to learn, so the models are biased to learn the shortcut in entity names while paying less attention to the context.

In this work, we systematically locate the origin of NER dataset bias and investigate its effect on model robustness from a dataset difficulty view. We try to answer two questions:

**Q1: *How does the entity-context distribution in the training data induce dataset bias?*** To answer this question, we borrow a recent concept "$\mathcal{V}$-information" (Xu et al., 2019; Ethayarajh et al., 2022) to measure the difficulty of entity and context for the model to learn in each dataset. We find that (1) In all NER datasets we examine, the $\mathcal{V}$-information of the entity is obviously larger than that of the context, indicating the dataset distribution induces models to learn more entity than context. (2) We further design an instance-level metric to measure the difficulty of entity and context in every single instance. We find that the largest population of instances—with equality-informative entity and context—does not lead or even harm the models to learn context.

**Q2: *Based on the analysis in Q1, are we able to build more robust NER datasets by altering the entity-context distribution in existing data?*** Based on Q1, we consider three potential ways to de-bias the NER datasets: (1) Reducing the overall $\mathcal{V}$-information of the entity in the training data. (2) Enhancing the overall $\mathcal{V}$-information of the context in the training data. (3) Enlarging the proportion of the robustness-helpful instances, i.e., instances with contexts easier than entities. By conducting extensive experiments, we verify the feasibility of all three approaches. These results also in turn confirm our analysis in **Q1**. Furthermore, we validate the transferability of the model-specific constructed datasets to improve the robustness of other models. These de-biased data are even helpful for existing model-based robustness-improving strategies, showing that building more

robust datasets is always fundamental for building more robust NER systems.

We hope our study can bring new insights into building more robust NER datasets, as well as developing more robust and general NER systems for real-world scenarios[1].

## 2 Measuring the Difficulty of Entity and Context in NER datasets

### 2.1 Background of $\mathcal{V}$-information

Recently, Xu et al. (2019) extends the mutual information Shannon (1948) to a concept of $\mathcal{V}$-**usable information** under computational constraints, which measures how much information about $Y$ can be extracted from $X$ with a certain model family $\mathcal{V}$. As defined in Xu et al. (2019):

**Definition 1.** *Let $X, Y$ be two random variables taking values in $\mathcal{X} \times \mathcal{Y}$. Let $\mathcal{V}$ be a predictive family that $\mathcal{V} \subseteq \Omega = \{f : \mathcal{X} \cup \{\varnothing\} \to P(\mathcal{Y})\}$. The $\mathcal{V}$-usable information is:*[2]

$$I_{\mathcal{V}}(X \to Y) = H_{\mathcal{V}}(Y|\varnothing) - H_{\mathcal{V}}(Y|X) \quad (1)$$

*where*

$$\begin{aligned} H_{\mathcal{V}}(Y|\varnothing) &= \inf_{f \in \mathcal{V}} \mathbb{E}[-\log_2 f[\varnothing](Y)] \\ H_{\mathcal{V}}(Y|X) &= \inf_{f \in \mathcal{V}} \mathbb{E}[-\log_2 f[X](Y)] \end{aligned} \quad (2)$$

More intuitively, $\mathcal{V}$ can be a pre-trained model family like BERT. $H_{\mathcal{V}}(Y|\varnothing)$ can be computed with a BERT model fine-tuned with a null input $\varnothing$ and $Y$, and $H_{\mathcal{V}}(Y|X)$ can be computed with a BERT model fine-tuned with $(X, Y)$.

Ethayarajh et al. (2022) further extends $\mathcal{V}$-information to measure dataset difficulty. Intuitively, a higher $I_{\mathcal{V}}(X \to Y)$ means $\mathcal{V}$ is able to extract more usable information from $X$ about $Y$, thus indicating an easier dataset for $\mathcal{V}$ to learn. They also propose to compare different attributes of $X$ by computing $I_{\mathcal{V}}(\tau(X) \to Y)$, where $\tau(\cdot)$ is a transformation on $X$ to isolate an attribute $a$. For instance, we can transform the regular NLI (Bowman et al., 2015) inputs into hypothesis-only inputs to measure the $\mathcal{V}$-information of the hypothesis attribute.

Ethayarajh et al. (2022) also propose a new measure based on $\mathcal{V}$-information to measure pointwise difficulty, which refers to **pointwise $\mathcal{V}$-information (PVI)**:

---

[1]Our code is available at https://github.com/rtmaww/NERDataBias

[2]We use $\log_2$ to measure the entropies in bits of information following (Ethayarajh et al., 2022).

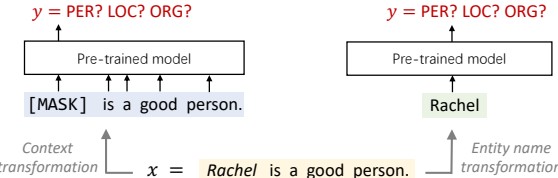

Figure 2: Illustration of our method to decouple contexts and entity names. We respectively train a context-only model and an entity-only model to examine the usable-information of each attribute.

**Definition 2.** *Let $X$, $Y$ be two random variables and $\mathcal{V}$ be a predictive family, the pointwise $\mathcal{V}$-information (PVI) of an instance $(x, y)$ is*

$$PVI(x \to y) = -\log_2 g[\varnothing](y) + \log_2 g'[x](y) \quad (3)$$

*where $g, g' \in \mathcal{V}$.*

Similarly, we can use a BERT model fine-tuned with $(X, Y)$ and a BERT model fine-tuned with $(\varnothing, Y)$ to calculate the PVI of every single instance, and a higher PVI indicates the instance is easier for $\mathcal{V}$. We adopt PVI to measure instance difficulty.

### 2.2 Decoupling Context and Entity

Motivated by Ethayarajh et al. (2022), in this work, we propose to decouple the entity and context attributes in order to measure the difficulty of each part respectively. Specifically, we first transform the NER dataset into two separate entity-only and context-only datasets. As shown in Fig.2, to build the context-only dataset, we replace the entity with a "[MASK]". To build the entity-only dataset, we simply use the entity as input. Then, we respectively train a context-only and an entity-only classification model based on a pre-trained model family (such as BERT) to predict the entity type based on the inputs. Based on the trained models, we can thus calculate the respective PVI of context and entity in each instance, as well as calculate the $\mathcal{V}$-information of context and entity of the whole training data, which indicates the difficulty of context and entity in this dataset to the used pre-trained model. More implementation details are included in Appendix A.3.

### 2.3 Context-entity Information Margin (CEIM)

In order to better describe the difficulty discrepancy of context and entity in an instance, we further introduce a new measure: Context-entity Information Margin (CEIM). Formally, we refer to the context-only model as $\mathcal{M}_C$, and the entity-only

model as $\mathcal{M}_E$. For each instance $(x, y)$, we denote the context PVI (measured by $\mathcal{M}_C$) as $\text{PVI}_C(x)$, and the entity PVI (measured by $\mathcal{M}_E$) as $\text{PVI}_E(x)$. The CEIM is then calculated by:

$$\text{CEIM}(x) = \text{PVI}_E(x) - \text{PVI}_C(x) \quad (4)$$

Intuitively, a high CEIM means the entity name in the instance is much easier to learn than the context.

## 3 How Does the Entity-context Distribution Induce Dataset Bias?

In order to answer **Q1**, in this section, we decouple the context and entity in NER datasets and calculate the $\mathcal{V}$-information of each part, so as to obtain the context-entity difficulty distribution. Based on the results, we analyze the correlation between a context-entity distribution and dataset bias in NER datasets at both the whole-data level (Sec.3.2) and the instance level (Sec.3.3).

### 3.1 Experiment Setup

**Dataset** To obtain comprehensive knowledge of the entity-context distribution of different NER datasets, we first conduct experiments on 6 commonly used datasets to calculate the $\mathcal{V}$-information of each dataset: CoNLL 2003 (Tjong Kim Sang and De Meulder, 2003), OntoNotes 5.0 (Weischedel et al., 2013), Bio-NER (Collier and Kim, 2004), ACE 2005 (Walker et al., 2006), Twitter (Zhang et al., 2018) and WNUT 2017 (Derczynski et al., 2017).

In Section 3.3, we include experiments to further explore the correlation between the entity-context distribution and model robustness. Following previous works on NER robustness (Lin et al., 2020; Wang et al., 2022), we experiment on CoNLL 2003 and ACE 2005 datasets. Except for evaluating standard performance on the i.i.d. test set (denoted as **Test**), we adopt two robustness test sets (Wang et al., 2021) for each dataset, the **OOV** and **CrossCategory** test sets, as the measure of model robustness, also following (Wang et al., 2022). These two robustness test set corresponds to the two typical failures of NER models as described in Fig.1, respectively. We include more dataset details and example cases of these test sets in the Appendix A.2.

**Base Model** We conducted all experiments in this section based on the BERT-base-cased pre-trained model (Devlin et al., 2019). More implementation details are included in Appendix A.3.

## 3.2 $\mathcal{V}$-information Comparison between Context and Entity

In Figure 3, we show the isolated $\mathcal{V}$-information of context and entity in 6 common NER datasets. From the results, we can observe that (1) In most of the commonly-used datasets, the $\mathcal{V}$-information of the entity is larger than 1.0. Such high values indicate the pre-trained model is able to learn and correctly classify most entities in the datasets without any information from the context. (2) In all datasets, the $\mathcal{V}$-information of the entity is obviously higher than that of context, meaning that the entity is much easier for the pre-trained model to learn than the context. Such $\mathcal{V}$-information discrepancy in existing datasets means that the difficulty distribution of existing datasets induces the model to learn more from entities instead of contexts. Corresponding to previous studies (Agarwal et al., 2020; Kim and Kang, 2022; Lin et al., 2020), this is an intrinsic bias in NER datasets that harms model robustness.

## 3.3 Understanding Entity-context Distribution with CEIM

As $\mathcal{V}$-information is the difficulty measure of the whole dataset, we step further to understand the instance-level difficulty of context and entity with Context-entity Information Margin (CEIM) (Section 2.3). We calculate the CEIM of instances in each dataset and divide the instances into 3 categories: (1) **High-CEIM**: instances with high CEIM, i.e., the entity is easier to learn than the context; (2) **Low-CEIM**: instances with low CEIM, i.e., the context is easier to learn; (3) **NZ-CEIM**: instances with a near-zero CEIM ($|\text{CEIM}(x)| < 0.5$), i.e., the context and entity is equally-easy to learn.

Table 1 shows the distribution of different CEIM instances and the average $\text{PVI}_E$ in each part. It's shown that the near-zero-CEIM instances cover the largest proportion in all datasets, and next is the high-CEIM instances. Also, the Average-$\text{PVI}_E$ shows that both the near-zero-CEIM and high-CEIM instances mainly contain high-PVI (easy-to-learn) entities.

In Table 2, we show some cases of different CEIM instances. Generally speaking, the high-CEIM instances often contain low-PVI contexts that are ambiguous or misleading, while containing informative entities. The near-zero-CEIM instances often consist of equally informative

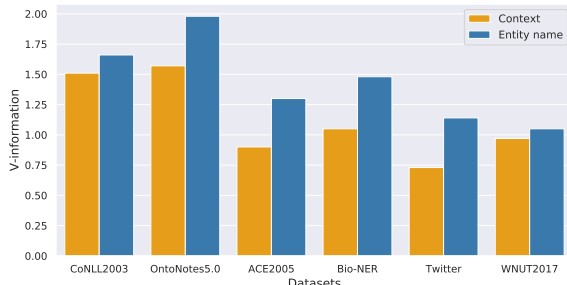

Figure 3: The $\mathcal{V}$-information of context and entity in 6 commonly used NER datasets.

contexts and entities. The low-CEIM instances consist of high-PVI context and low-PVI entities, where the entities are often rare, more complicated, or ambiguous. We include more discussion on how different CEIM instances might affect model prediction in Appendix A.1.

| Dataset | Low-CEIM | | NZ-CEIM | | High-CEIM | |
|---|---|---|---|---|---|---|
| | # num | A-PVI$_E$ | # num | A-PVI$_E$ | # num | A-PVI$_E$ |
| **CoNLL 2003** | 1962 | -1.01 | **18266** | 1.90 | 3271 | 1.95 |
| **OntoNotes 5.0** | 4694 | -1.55 | **37660** | 2.20 | 13390 | 2.59 |
| **ACE2005** | 3843 | -0.36 | **17337** | 1.51 | 5290 | 1.99 |
| **Twitter** | 957 | -1.34 | **3130** | 1.52 | 2089 | 1.64 |
| **WNUT2017** | 516 | -2.05 | **880** | 1.73 | 579 | 2.12 |
| **Bio-NER** | 1936 | -1.64 | **36774** | 1.38 | 12480 | 2.31 |

Table 1: The number of instances with different CEIM levels in each dataset. A-PVI$_E$ denotes the Average-PVI of entity names in different parts of data.

**How do different CEIM instances affect model robustness?** To further understand the impact of entity-context distribution on the model robustness, we further conducted an experiment to compare the model behavior when trained on data with different CEIMs. We first randomly sampled 1000 instances from the whole dataset as a baseline training set, denoted as ***Base***. Next, we randomly sampled 1000 near-zero-CEIM instances and add them to ***Base*** to construct a new 2000-instance training set, denoted as ***Base+NZ***. Similarly, we constructed another two training sets ***Base+High*** and ***Base+Low*** by adding 1000 randomly-sampled high-CEIM and low-CEIM instances to ***Base***, respectively. We then train a model on each training set and evaluate their performance on the test set and the robustness OOV and CrossCategory test set.

Table 3 shows the effect of different CEIM data on the model performance. We can see that: (1) By adding 1000 extra instances to Base, the performance on the test set is improved on all training sets. Among all training sets, Base+High

| Type | Context | Entity name | Label | $PVI_C$ | $PVI_E$ | CEIM |
|------|---------|-------------|-------|---------|---------|------|
| **High** | It's composed of the U.S., Russia, [MASK], and the United Nations. | the European Union | ORG | -4.73 | 2.04 | 6.77 |
| **CEIM** | [MASK] is playing a trick on us he says . | America | GPE | -5.25 | 1.78 | 7.03 |
| **Near-zero** | And equally without doubt, [MASK] did not want him, that is, Bush, to do this. | Blair | PERSON | 1.87 | 1.84 | -0.03 |
| **CEIM** | China and [MASK] have become important mutual trade partners. | Russia | GPE | 1.82 | 1.85 | 0.03 |
| **Low** | On Tuesday; [MASK] directors announced plans to spin off two big divisions... | Trelleborg 's | ORG | 2.19 | -0.12 | -2.07 |
| **CEIM** | ... Mr.[MASK] and I did the other night on ABC's " Nightline. " | Apple | PERSON | 1.63 | -6.62 | -8.25 |

Table 2: Examples of different CEIM instances. More detailed discussion can be found in Appendix A.1.

| CoNLL 2003 | | | |
|------|------|------|------|
| **Training set** | **Test** | **OOV** | **CrossCategory** |
| **Base** | 87.15 | 64.28 | 40.74 |
| **Base + High** | **87.61 (+0.46)** | 63.77 (-0.51) | 40.91 (+0.17) |
| **Base + NZ** | 87.51 (+0.36) | 64.05 (-0.23) | 41.55 (+0.81) |
| **Base + Low** | 87.50 (+0.35) | **67.96 (+3.68)** | **44.93 (+4.19)** |
| ACE 2005 | | | |
| **Training set** | **Test** | **OOV** | **CrossCategory** |
| **Base** | 81.44 | 73.91 | 39.04 |
| **Base + High** | 83.56 (+2.12) | 73.64 (-0.27) | 40.80 (+1.76) |
| **Base + NZ** | **84.74 (+3.30)** | 74.54 (+0.63) | 41.99 (+2.95) |
| **Base + Low** | 83.36 (+2.13) | **76.56 (+2.65)** | **42.86 (+3.82)** |

Table 3: The effect of different CEIM data on the model generalization and robustness.

and Base+NZ show slightly larger improvement than Base+Low, which corresponds to the findings in previous studies (Lin et al., 2020; Agarwal et al., 2020; Zeng et al., 2020) that high-PVI entity names contribute more to the generalization on the i.i.d. test set. (2) Base+Low shows notable improvement on both OOV and CrossCategory test sets, while Base+High and Base+NZ are less beneficial or even harmful to the robustness performance. These results demonstrate that *the low-CEIM instances, i.e., instances with contexts easier than entities, contribute most to the model robustness.* (3) The limited robustness performance of Base+NZ also indicates that although the context may be informative enough for label predicting, with equally easy entity and context, the pre-trained model still tends to make use of the entity information instead of context. Unfortunately, the near-zero-CEIM data constitutes the largest proportion in all datasets (Tab.1), thus having much larger effects on the model training than the low-CEIM data (which largely improves robustness).

## 4 Can We Build More Robust Datasets by Altering Entity-context Distribution?

Based on the above analysis, in this section, we explore three potential ways to de-bias NER datasets by altering the entity-context distribution.

### 4.1 Experiment Setup

**Datasets** Similar to Section 3, in this section, we conduct experiments on the CoNLL2003 and ACE2005 datasets, and evaluate the i.i.d. test performance on the Test set, as well as the robustness performance on the OOV and CrossCategory test sets (denoted as "Cate.").

**Base Model and Baselines** In this section, we conduct all experiments based on two pre-trained LM, BERT-base-cased (Devlin et al., 2019) and RoBERTa-large (Liu et al., 2019). To better verify the robustness improvement of the three de-bias methods, we include several robustness-improving baselines: (1) *Base* (Devlin et al., 2019; Liu et al., 2019) The base token classification model based on BERT-base-cased and RoBERTa-large, trained on the original training sets; (2) *DataAug* (Dai and Adel, 2020), which augments the training set by replacing entities with similar entities or typos entities; (3) *MINER* (Wang et al., 2022), which also creates samples with entity switching and trained with a contrastive robustness-improving loss. It is also the SOTA method in NER robustness; (4) *LPFT* (Kumar et al., 2022), a general OOD method that can effectively improve OOD model generalization. More implementation details can be found in Appendix A.4.

### 4.2 Enlarging the Low-CEIM Proportion

The experiments in Section 3.3 have shown that the high-CEIM data and near-zero-CEIM data contribute less to the model robustness, while the low-CEIM data do make the model learn more from context. However, the proportion of the low-CEIM data in the datasets is quite low, leading to limited influence on the model learning. In this section, we reconstruct the training sets to alter the proportion of low-CEIM data, trying to investigate: (1) If enlarging the proportion of low-CEIM data is a feasible way to improve robustness; (2) If we can achieve a good balance of robustness and generalization with a certain proportion of low-

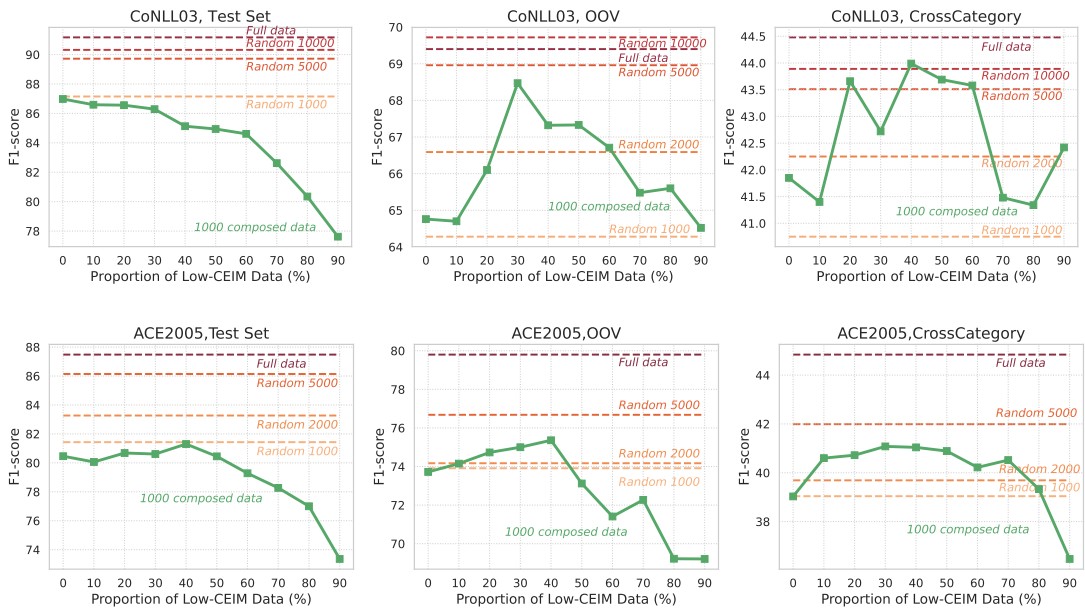

Figure 4: Performance of the BERT-base-cased models trained on the reconstructed datasets of different low-CEIM proportions. We report the average result of 3 repeated experiments. We also include the results of the RoBERTa-large models in Appendix A.5.

CEIM data.

### 4.2.1 Detailed Experimental Settings

As the number of low-CEIM instances is limited in both CoNLL2003 and ACE2005 datasets (Table 1), we decided to fix the instance number of the reconstructed training sets to 1000. We construct datasets with the proportion of low-CEIM data ranging from 0% to 90%. For example, when constructing a training set with 20% low-CEIM data, we randomly sample 200 instances from the low-CEIM data and 800 instances from the high- or near-zero-CEIM data.

### 4.2.2 Results

Figure 4 shows the model performance trained on the reconstructed training sets on BERT-base-cased [3]. We can observe that: (1) With the proportion of low-CEIM data increasing, the performance on the test set keeps dropping, which corresponds to the discussion in Sec.3.3 that the high- and near-zero-CEIM data contribute more to i.i.d. generalization. (2) As the proportion of low-CEIM data increases, the performance on the OOV and CrossCategory test sets generally shows a trend of first rising and then falling. The rising robustness performance at low low-CEIM proportion validates the effectiveness of enlarging the low-CEIM proportion for improving model robustness. However, as the low-CEIM proportion

keeps increasing, the poor generalization ability will also affect robustness and lead to declining results. (3) Compared with the results on 1000 random data, the reconstructed data, with certain data proportions, achieves significant improvement on the robustness test sets and comparable performance on the Test set. We conclude that *for different datasets, there exists appropriate proportions when a good balance of i.i.d. and OOD generalization can be achieved* (e.g., 10%-30% for CoNLL03, 20%-40% for ACE05).

### 4.3 Reducing the $\mathcal{V}$-information of Entity

As discussed in Section 3.2, the $\mathcal{V}$-information discrepancy between entity and context is the main factor of dataset bias in NER. In order to de-bias NER datasets, the most intuitive idea is to increase the low-PVI entities or decrease the high-PVI entities in the training set. In this section, we explore potential ways to reduce the $\mathcal{V}$-information of the entity of the training set by dataset reconstruction:

**A. Random2low:** Increasing the low-PVI entities by randomly replacing high-PVI entities with low-PVI entities. The replaced sentences are added to the original dataset as data augmentation.

**B. HighC2low:** Based on A, we further select instances with high-PVI context and high-PVI entity for entity replacement. This process

---

[3]The results on RoBERTa-large are in Appendix A.5.

| Methods | | Base Model: BERT-base-cased | | | | | | | | Base Model: RoBERTa-large | | | | | | | |
|---|---|---|---|---|---|---|---|---|---|---|---|---|---|---|---|---|---|
| | | CoNLL2003 | | | | ACE2005 | | | | CoNLL2003 | | | | ACE2005 | | | |
| | | Test | OOV | Cate. | Avg. | Test | OOV | Cate. | Avg. | Test | OOV | Cate. | Avg. | Test | OOV | Cate. | Avg. |
| Base | | 91.18 | 70.89 | 44.48 | 68.85 | 87.48 | 79.80 | 44.84 | 70.71 | 92.10 | 72.54 | 50.01 | 71.55 | 88.84 | 80.84 | 46.13 | 71.94 |
| DataAug | | 90.52 | 73.45 | 46.39 | 70.12 | 87.16 | 80.99 | 44.58 | 70.91 | 91.71 | 76.93 | 52.18 | 73.60 | 88.96 | 81.95 | 46.03 | 72.31 |
| LPFT | | 91.14 | 73.89 | 48.32 | 71.12 | 87.76 | 80.28 | 44.37 | 70.80 | 92.12 | 75.67 | 48.53 | 72.11 | 88.85 | 82.10 | 46.23 | 72.39 |
| MINER | | 90.98 | 76.89 | 49.97 | 72.61 | 87.41 | 79.23 | 45.45 | 70.70 | 91.88 | 79.33 | 56.27 | 75.83 | 88.83 | 82.69 | 46.75 | 72.76 |
| A.Random2low | 20% | 90.84 | 76.04 | 48.24 | 71.71 | 87.51 | 81.23 | 47.25 | 71.99 | 91.84 | 78.05 | 54.95 | 74.95 | 88.23 | 82.69 | 52.13 | 74.35 |
| | 40% | 90.42 | 76.03 | 47.72 | 71.39 | 87.54 | 81.64 | 48.28 | 72.49 | 91.99 | 79.00 | 56.78 | 75.93 | 88.54 | 81.88 | 51.24 | 73.88 |
| B.HighC2Low | 20% | 91.19 | 75.84 | 48.45 | 71.83 | 87.43 | 82.18 | 48.76 | 72.79 | 92.18 | 78.36 | 56.11 | 75.55 | 88.56 | 82.81 | 51.90 | 74.42 |
| | 40% | 90.55 | 77.11 | 48.92 | 72.19 | 87.11 | 82.32 | 49.97 | 73.13 | 91.76 | 79.37 | 57.17 | 76.10 | 88.42 | 83.55 | 52.72 | 74.90 |
| C.Redundant2Low | 20% | 90.71 | 76.35 | 48.02 | 71.69 | 87.37 | 81.28 | 48.59 | 72.41 | 91.82 | 79.23 | 57.11 | 76.05 | 88.66 | 84.07 | 51.89 | 74.87 |
| | 40% | 90.42 | 77.34 | 48.07 | 71.94 | 86.97 | 82.17 | 48.41 | 72.52 | 91.54 | 79.52 | 58.06 | 76.38 | 88.67 | 84.43 | 53.49 | 75.53 |
| *Dataset Transferability Study* (Discussed in Section 4.5) | | | | | | | | | | | | | | | | | |
| MINER + HighC2Low 40% | | 90.71 | 79.85 | 53.42 | 74.66 | 87.01 | 80.75 | 49.87 | 72.55 | 91.70 | 80.33 | 61.16 | 77.73 | 89.37 | 85.16 | 54.54 | 76.36 |
| BERT.HighC2Low 40% | | – | – | – | – | – | – | – | – | 91.74 | 79.18 | 55.15 | 75.36 | 89.11 | 83.37 | 52.23 | 74.90 |

Table 4: Performance of different approaches to reduce the $\mathcal{V}$-information of entity in the datasets. Each reported result is averaged by 3 repeated experiments. More detailed results of different rates are reported in Appendix A.5.

ensures that the replaced instances have informative contexts for the model to learn and predict.

**C. Redundant2low:** As declared in (Lin et al., 2020), existing datasets mainly consist of regularly-patterned entities that harm model robustness. We also find in our experiments that there exists a large number of redundant high-PVI entities in the datasets. Therefore, we propose to reduce these redundant entities by replacing them with low-PVI entities. Different from A and B, this method doesn't increase the total data size.

*Note that all methods are actually trying to introduce more low-CEIM instances.* In our experiments, we reconstruct the existing datasets based on the above methods and train new models on the reconstructed training sets, separately.

### 4.3.1 Results

Table 4 shows the results of the models trained on three types of reconstructed datasets. Here, $m\%$ means replacing $m\%$ of the total entities in each method. From the results, we can observe that: (1) All of the three methods show obvious improvement over Base and even DataAug on OOV and CrossCategory. As the replacement rate grows, the robustness performance generally increases, as well as the test performance decreases. This trend is similar to previously observed trends in Fig.4. (2) Compared with A.Random2Low, B.HighCLow shows relatively higher robustness performance and test performance. This is because randomly replacing high-PVI entities

with low-PVI entities without considering the context might create instances with both difficult entity and context, which would not benefit robustness and generalization. In contrast, B.HighC2Low ensures introducing more low-CEIM instances. In most cases, B.HighC2Low can construct a dataset that ***achieves both higher robustness performance and comparable test performance***. (3) C.Redundant2Low is also effective in improving model robustness. Although a large proportion of high-PVI entities are replaced, the test performance is only slightly hurt and comparable to A.Random2Low and DataAug that increases data size with augmentation. It also achieves larger improvement on the robustness test sets than A.Random2Low and DataAug. These results demonstrate that the large number of redundant high-PVI entities in the NER datasets limitedly benefits model generalization yet would harm model robustness. (4) Apart from the three methods, MINER can also achieve good robustness. Nevertheless, we claim that the dataset reconstruction methods are generally orthogonal to the model-based robustness-improving methods like MINER. More details of this point are discussed in Sec.4.5.

### 4.4 Enhancing the $\mathcal{V}$-information of Context

Aside from lowering the $\mathcal{V}$-information of the entity in Sec.4.3, another intuitive approach is to enhance the $\mathcal{V}$-information of the context. Similar to Sec.4.3, in this section, we design two potential ways to increase the proportion of high-PVI

| Methods | | Base Model: BERT-base-cased | | | | | | | | Base Model: RoBERTa-large | | | | | | | |
| --- | --- | --- | --- | --- | --- | --- | --- | --- | --- | --- | --- | --- | --- | --- | --- | --- | --- |
| | | CoNLL2003 | | | | ACE2005 | | | | CoNLL2003 | | | | ACE2005 | | | |
| | | Test | OOV | Cate. | Avg. | Test | OOV | Cate. | Avg. | Test | OOV | Cate. | Avg. | Test | OOV | Cate. | Avg. |
| Base | | **91.18** | 70.89 | 44.48 | 68.85 | 87.48 | 79.80 | 44.84 | 70.71 | 92.10 | 72.54 | 50.01 | 71.55 | 88.84 | 80.84 | 46.13 | 71.94 |
| DataAug | | 90.52 | 73.45 | 46.39 | 70.12 | 87.16 | 80.99 | 44.58 | 70.91 | 91.71 | 76.93 | 52.18 | 73.60 | 88.96 | 81.95 | 46.03 | 72.31 |
| LPFT | | 91.14 | 73.89 | 48.32 | 71.12 | **87.76** | 80.28 | 44.37 | 70.80 | **92.12** | 75.67 | 48.53 | 72.11 | 88.85 | 82.10 | 46.23 | 72.39 |
| MINER | | 90.98 | 76.89 | 49.97 | 72.61 | 87.41 | 79.23 | 45.45 | 70.70 | 91.88 | 79.33 | 56.27 | 75.83 | 88.83 | 82.69 | 46.75 | 72.76 |
| I.Random2High | 30% | 90.95 | 75.31 | 48.27 | 71.51 | 86.74 | 80.70 | 45.03 | 70.83 | 91.88 | 77.66 | 54.52 | 74.69 | 88.67 | 84.66 | 47.70 | 73.67 |
| | 40% | 90.43 | 76.69 | 50.20 | 72.44 | 86.87 | 81.21 | 45.26 | 71.12 | 91.67 | 77.86 | 54.85 | 74.79 | 88.53 | 84.08 | 48.81 | 73.81 |
| II.Low2High | 30% | 90.42 | 76.87 | 52.15 | 73.15 | 86.99 | 81.38 | 46.78 | 71.71 | 90.76 | 79.12 | 60.52 | 76.80 | 88.86 | 83.60 | 50.57 | 74.34 |
| | 40% | 89.72 | 78.41 | 55.70 | 74.61 | 86.48 | 82.01 | 47.78 | **72.09** | 90.92 | 79.58 | 61.93 | 77.47 | 88.67 | 84.40 | **52.20** | **75.09** |
| Dataset Transferability Study (Discussed in Section 4.5) | | | | | | | | | | | | | | | | | |
| MINER + Low2High 40% | | 90.18 | **79.69** | 56.03 | **75.30** | 87.13 | 80.16 | **48.72** | 72.00 | 91.72 | **80.05** | 61.03 | **77.60** | 89.33 | 84.67 | 50.07 | 74.69 |
| BERT.Low2High 40% | | – | – | – | – | – | – | – | – | 91.05 | 79.63 | 59.79 | 76.82 | 88.79 | 83.28 | 51.88 | 74.65 |

Table 5: Performance of different approaches to enhance the $\mathcal{V}$-information of context in the datasets. Each reported result is averaged by 3 repeated experiments. More detailed results of different rates are reported in Appendix A.5.

contexts in the training sets:

**I.Random2High**   Randomly deleting a certain ratio of contexts and replacing them with high-PVI contexts from the retained set.

**II.Low2High**   Deleting a certain ratio of low-PVI contexts, and replacing the deleted contexts with high-PVI contexts.

Similar to Section 4.3, we reconstruct the existing datasets based on the above methods and train new models on the reconstructed data, separately.

### 4.4.1   Results

Table 5 shows the results of different methods, where $m\%$ means deleting (replacing) $m\%$ of the total context. We can observe that: (1) Randomly deleting a certain proportion of context, although decreases context diversity in the datasets, doesn't show a serious drop in the test performance. Interestingly, by replacing these contexts with high-PVI contexts, the robustness performance can be improved to a certain degree. (2) As shown in Appendix A.1, low-PVI contexts are often noisy contexts and will harm performance. Therefore, it is intuitive that replacing low-PVI contexts with high-PVI contexts effectively improves robustness performance. It also outperforms I.Random2High on most replacement rates, validating that reducing context difficulty can largely improve model robustness. However, this method suffers from more decrease in the test performance, which might be because reducing the low-PVI contexts relatively reduces model attention on the entity,

thus hindering generalization on i.i.d. distribution. Nevertheless, the results show that a good trade-off between OOD and i.i.d. performance can be achieved with certain replacement ratios.

### 4.5   Transferability of the De-biased Datasets

As the $\mathcal{V}$-information calculation is model-specific, it is intuitive to wonder if the datasets reconstructed based on one model can transfer to another model. In both Table 4 and Table 5, we further conduct experiments to investigate the transferability of the de-biased datasets. In both experiments, we trained MINER on the reconstructed datasets (MINER+HighC2Low 40% and MINER+Low2High 40%). We also use the BERT-based reconstructed datasets to train RoBERTa-large models (BERT.HighC2Low 40% and BERT.Low2High 40%). Surprisingly, the reconstructed datasets can also benefit other models to a certain degree. These results not only validate the generalizability of the de-biased datasets, but also reveal that the dataset reconstruction methods are orthogonal to the model-based robustness-improving methods such as MINER. It also shows that ***building more robust datasets is fundamental for building more robust NER systems***.

## 5   Related Work

### 5.1   Analyzing the Robustness Problem in NER.

Many works have focused on analyzing the robustness problem in NER. These works generally fall into two categories: (1) Constructing challenging sets to evaluate the model robustness (Ghaddar

et al., 2021; Lin et al., 2020) such as switching the entities in test sets (Agarwal et al., 2020), testing on out-of-dictionary entities (Lin et al., 2020; Kim and Kang, 2022) or introducing typos to create OOV entities (Wang et al., 2021); (2) Investigating the impact of possible attribute through a delicate design of experiments (Fu et al., 2020; Kim and Kang, 2022), such as conducting randomization experiments (Lin et al., 2020) or measuring the impact of attributes with specifically-designed metrics (Fu et al., 2020). Our work is totally different from the existing studies. We provide a brand new view of considering the NER robustness problem by quantifying and comprehensively analyzing the correlation between context-entity distribution and model robustness and provide new insights into the NER robustness studies.

It is worth mentioning that Lin et al. (2020) and Peng et al. (2020) (work on Relation Extraction) also consider the "context or name" problem and design experiments to disentangle the impact of context and entity name. These experiments also motivate the experiment designs in this work.

## 5.2 Mitigating the Robustness Problem in NER.

There are also many works that aim at mitigating the robustness problem in NER. These works include methods to alleviate the OOV problem (Bojanowski et al., 2017; Peng et al., 2019), leveraging data augmentation (Dai and Adel, 2020; Zeng et al., 2020) or adopting adversarial training (Ghaddar et al., 2021) or other training strategies (Wang et al., 2022) to improve model robustness. In this work, we explore a new direction of approaches: to improve model robustness through data reconstruction. We also argue that constructing robust datasets is fundamental for building more robust NER systems.

## 6 Conclusion

In this work, we conduct an interesting study on the model robustness problem of NER. We quantify the difficulty distribution of context and entity in existing NER datasets and reveal how the entity-context distribution affects model robustness. Based on our analysis, we further explore three potential ways to de-bias the existing NER datasets by reconstructing the existing datasets and altering the entity-context distribution. With extensive experiments, we validate that the reconstructed

datasets can largely improve model robustness. As these datasets can also benefit model-based robustness-improving methods, we argue that building more robust datasets is fundamental for building more robust NER systems.

## Acknowledgement

The authors wish to thank the anonymous reviewers for their helpful comments. This work was partially funded by National Natural Science Foundation of China (No.62076069, 61976056) and Shanghai Academic Research Leader Program 22XD1401100.

## Limitations

We summarize the limitations of this work as follows: (1) We regard our work as an exploratory study on the robustness problem of NER from the dataset view. Nevertheless, the method to measure the dataset difficulty is not quite efficient since it requires first training two individual models on an i.i.d. training set. There might be more efficient ways to measure the dataset and instance difficulty, thus further improving the efficiency and practicality of the data reconstruction methods. (2) In this work, we consider the robustness problem of NER models with only small pre-trained models (model size less than 1B). As the large language models have shown powerful ability in information extraction tasks, it is in doubt that if the same conclusion can also generalize to large language models. Nevertheless, we believe our study is worthful since building more robust datasets is always important for building a NER system required for real-world usage. We expect our work can prompt more valuable future works also on large language models.

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

# A Appendix

## A.1 Discussion of Different CEIM instances

In Section 3.3, we categorized each dataset into three categories based on the CEIM scores: high-CEIM instances, low-CEIM instances, and Near-zero-CEIM instances. Then, we conducted experiments to show the effect of different CEIM instances on model robustness. To better understand the concept of CEIM, we provide some examples of different CEIM instances in Tab.2.

Generally, the high-CEIM instances often contain low-PVI contexts. These contexts are often ambiguous or misleading, thus might introduce noise in model learning. On the contrary, the entities in these high-CEIM instances are often informative enough for the model to predict. As a result, the model will pay less attention to the context when trained on high-CEIM instances. Therefore, removing these low-PVI contexts is also helpful for de-bias the datasets, as shown in Sec.4.4.

In near-zero-CEIM instances, the context and the entity are usually equally supportive (the number of cases with equally-low context and entity is small). However, the entity pattern is easier to learn and memorize. We deduce that the model might still tend to memorize entity patterns instead of learning more context, corresponding to the results in Table 3.

The low-CEIM instances consist of high-PVI context and low-PVI entities. These entities are often ambiguous, more complicated or less frequent entities. Therefore, trained on low-CEIM instances, the model will tend to pay attention to the informative context for label predicting, which will benefit robustness. However, as the generalization to the i.i.d. test sets mainly relies on high-PVI entities, these low-CEIM instances, although helpful in OOD situations, might harm i.i.d. generalizations. In Section 4.2,4.3,4.4, we validate that an appropriate context-entity distribution can achieve better trade-off between i.i.d. and OOD generalization.

## A.2 Dataset Details

In Section 3, we examine and analyze the $\mathcal{V}$-information of 6 commonly used datasets, including: CoNLL 2003 (Tjong Kim Sang and De Meulder, 2003) from the newswire domain,

OntoNotes 5.0 (Weischedel et al., 2013)[4] from the general domain, Bio-NER (Collier and Kim, 2004) from the biology domain, ACE 2005 (Walker et al., 2006)[5] the general domain, Twitter (Zhang et al., 2018) from social media domain and WNUT 2017 (Derczynski et al., 2017) from social media domain. The dataset details can be found in Table 6.

In Section 4 and further exploration in Section 3, we conduct experiments on CoNLL 2003 and ACE 2005 to investigate the relation between the entity-context distribution and model robustness. Following (Wang et al., 2022), we adopt two robustness test sets from (Wang et al., 2021): the OOV test set and the CrossCategory test set (denoted as "Cate."). The OOV test set is constructed by replacing entities with out-of-vocabulary entities of the same category. The CrossCategory test set is constructed by replacing entities with entities from different categories. Therefore, these two robustness test sets respectively correspond to the two typical failures of SOTA NER models shown in Figure 1. In Table 9, we show some cases of these two robustness test sets for better understanding. We also include quantity details of these two test sets in Table 6.

## A.3 Implementation Details for $\mathcal{V}$-information Experiments

In Section 3, we conducted experiments to calculate the $\mathcal{V}$-information and PVIs of each dataset. For both context-only and entity-only models, we implemented based on the text classification model based on the Huggingface library[6]. To measure the PVI of the whole training set, we adopted 5-fold cross-validation. As the PVI of an instance $(x, y)$ only depends on the distribution of $(X, Y)$, using 5-fold cross-validation will not affect the estimation of PVI (Ethayarajh et al., 2022). We detail the hyperparameters used for the experiments in Table 7. Other hyperparameters are the same as the default hyperparameters if not noted.

All experiments are conducted on NVIDIA GeForce RTX 3090 and NVIDIA Tesla V100. For each result, we report the average results of 3 repeated experiments.

---

[4]https://catalog.ldc.upenn.edu/license/ldc-non-members-agreement.pdf

[5]https://catalog.ldc.upenn.edu/license/ldc-non-members-agreement.pdf

[6]https://github.com/huggingface/transformers/tree/main/examples/pytorch/text-classification

| Datasets | Domain | Language | # Class | # Train | # Dev | # Test | # OOV | # Cate. |
|----------|--------|----------|---------|---------|-------|--------|-------|---------|
| CoNLL 2003 | News | English | 4 | 15.0k | 3.5k | 3.5k | 3.5k | 3.5k |
| OntoNotes 5.0 | General | English | 18 | 60.0k | 8.5k | 8.3k | – | – |
| ACE 2005 | General | English | 8 | 9.7k | 2.3k | 2.0k | 1.3k | 1.3k |
| WNUT 2017 | Social media | English | 6 | 3.4k | 1.0k | 1.3k | – | – |
| Twitter | Social media | English | 4 | 4.0k | 1.0k | 3.3k | – | – |
| BioNER | Biology | English | 5 | 18.5k | 0.1k | 3.9k | – | – |

Table 6: Dataset details.

| Dataset | Model | Epoch | lr | Batch size |
|---------|-------|-------|-----|------------|
| \multicolumn{5}{Base model: BERT-base-cased} | | | | |
| CoNLL2003 | Context-only | 2 | 5e-5 | 32 |
| | Entity-only | 1/2 | 5e-5 | 32 |
| OntoNotes 5.0 | Context-only | 2 | 5e-5 | 32 |
| | Entity-only | 1/2 | 5e-5 | 32 |
| ACE2005 | Context-only | 2 | 5e-5 | 32 |
| | Entity-only | 1/2 | 5e-5 | 32 |
| WNUT | Context-only | 5 | 1e-4 | 32 |
| | Entity-only | 3 | 1e-4 | 32 |
| Twitter | Context-only | 5 | 1e-4 | 32 |
| | Entity-only | 3 | 1e-4 | 32 |
| BioNER | Context-only | 3 | 5e-5 | 32 |
| | Entity-only | 1 | 5e-5 | 32 |
| \multicolumn{5}{Base model: RoBERTa-large} | | | | |
| ACE2005 | Context-only | 2 | 1e-5 | 32 |
| | Entity-only | 1/2 | 1e-5 | 16 |
| CoNLL2003 | Context-only | 2 | 1e-5 | 32 |
| | Entity-only | 1/2 | 1e-5 | 16 |

Table 7: Hyperparameters used for $\mathcal{V}$-information experiments in Section 3.

## A.4 Implementation Details for Dataset Reconstruction Experiments

We implemented all BERT-base-cased and RoBERTa-large dataset reconstruction experiments based on the token classification model in the Huggingface library[7]. For DataAug, we implemented based on the implementations from (Wang et al., 2022)[8]. For MINER, we implemented based on the released code[9]. We used batch_size=32, gama=0.001, beta=0.01, lr=0.00001 for BERT-based baselines, and used batch_size=32, gama=0.0001, beta=0.0001, lr=0.00001 for RoBERTa-based baselines. For LPFT, we also implemented based on the token classification model in the Huggingface library[10]. We include the hyperparameters used for the experiments in Table 8. Other hyperparameters

are the same as the default hyperparameters if not noted.

All experiments are conducted on NVIDIA GeForce RTX 3090 and NVIDIA Tesla V100. For each result, we report the average results of 3 repeated experiments.

| Dataset | Training set size | Epoch | lr | Batch size |
|---------|-------------------|-------|-----|------------|
| \multicolumn{5}{Base model: BERT-base-cased} | | | | |
| CoNLL2003 | <5000 | 10 | 5e-5 | 8 |
| | >5000 | 7 | 5e-5 | 16 |
| ACE2005 | <5000 | 10 | 5e-5 | 8 |
| | >5000 | 10 | 5e-5 | 16 |
| \multicolumn{5}{Base model: RoBERTa-large} | | | | |
| CoNLL2003 | <5000 | 10 | 1e-5 | 8 |
| | >5000 | 5 | 1e-5 | 16 |
| ACE2005 | <5000 | 10 | 1e-5 | 8 |
| | >5000 | 5 | 1e-5 | 16 |

Table 8: Hyperparameters used for all dataset reconstruction experiments, DataAug, and LPFT baselines in Section 4. Also used for the experiments in Section 3.3. For LPFT, we train LP models and LPFT models with the same hyperparameters as in the table.

## A.5 Detailed Experiment Results of Data Reconstruction

In Section 4, we only showed a part of the experiment results due to space limitation. In this section, we include other detailed experiment results, including the results based on RoBERTa-large in Section 4.2 (Fig.5), the detailed results of different data reconstruction rates in Section 4.3 (Table 10) and Section 4.4 (Table 11).

---

[7]https://github.com/huggingface/transformers/tree/main/examples/pytorch/token-classification
[8]https://github.com/BeyonderXX/MINER
[9]https://github.com/BeyonderXX/MINER
[10]https://github.com/huggingface/transformers/tree/main/examples/pytorch/token-classification

| Dataset | Test set | Example |
|---------|----------|---------|
| CoNLL2003 | OOV | Bazemore (PER) said the Commodore Books (ORG) 's credibility was at stake over the issue of trade and labour . |
| CoNLL2003 | Cate. | The Yellow (PER) missed his club 's last two games after the " National " (ORG) slapped a worldwide ban on him... |
| ACE2005 | OOV | Duygu (PER) happen to be at a very nice Ginowan (LOC) by the beach Lotte Capital (LOC) this is a chance for Evon (PER) to get away from Sacred Heart Secondary school (ORG) coverage , everything , and kind of relax |
| ACE2005 | Cate. | Reporter : Solomon (PER) is associated with Clive (ORG) since 1987 and been very successful . |

Table 9: Examples of the robustness test sets. Cate. denotes the CrossCategory test set.

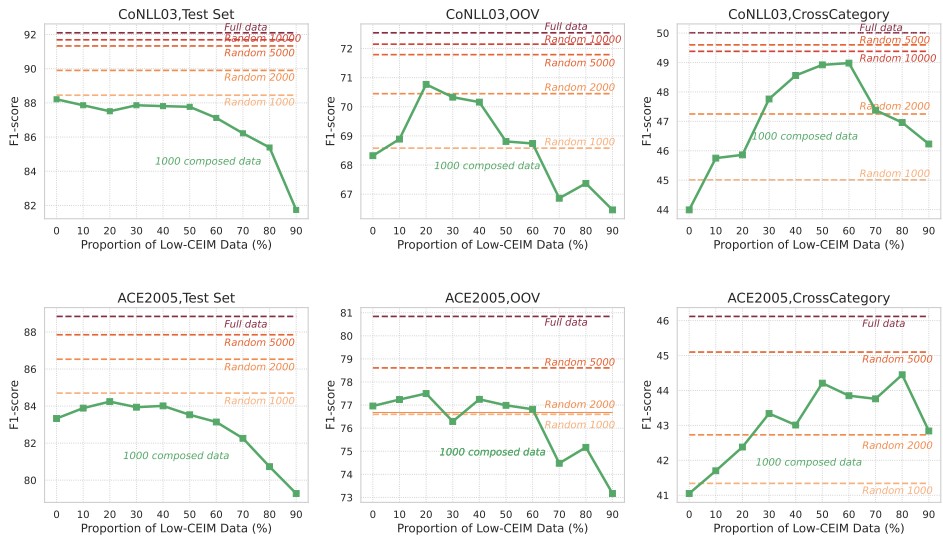

Figure 5: Performance of the RoBERTa-large models trained on the reconstructed datasets of different low-CEIM proportions. We report the average result of 3 repeated experiments.

| Methods | | Base Model: BERT-base-cased | | | | | | | | Base Model: RoBERTa-large | | | | | | | |
| | | CoNLL2003 | | | | ACE2005 | | | | CoNLL2003 | | | | ACE2005 | | | |
| | | Test | OOV | Cate. | Avg. | Test | OOV | Cate. | Avg. | Test | OOV | Cate. | Avg. | Test | OOV | Cate. | Avg. |
|---|---|---|---|---|---|---|---|---|---|---|---|---|---|---|---|---|---|
| **Base** | | 91.18 | 70.89 | 44.48 | 68.85 | 87.48 | 79.80 | 44.84 | 70.71 | 92.10 | 72.54 | 50.01 | 71.55 | 88.84 | 80.84 | 46.13 | 71.94 |
| **DataAug** | | 90.52 | 73.45 | 46.39 | 70.12 | 87.16 | 80.99 | 44.58 | 70.91 | 91.71 | 76.93 | 52.18 | 73.60 | 88.96 | 81.95 | 46.03 | 72.31 |
| **LPFT** | | 91.14 | 73.89 | 48.32 | 71.12 | **87.76** | 80.28 | 44.37 | 70.80 | 92.12 | 75.67 | 48.53 | 72.11 | 88.85 | 82.10 | 46.23 | 72.39 |
| **MINER** | | 90.98 | 76.89 | 49.97 | 72.61 | 87.41 | 79.23 | 45.45 | 70.70 | 91.88 | 79.33 | 56.27 | 75.83 | 88.83 | 82.69 | 46.75 | 72.76 |
| **A.Random2low** | 10% | 90.99 | 73.67 | 46.17 | 70.28 | 87.24 | 80.43 | 46.69 | 71.45 | 91.97 | 76.59 | 53.65 | 74.07 | 88.85 | 81.98 | 49.43 | 73.42 |
| | 20% | 90.84 | 76.04 | 48.24 | 71.71 | 87.51 | 81.23 | 47.25 | 71.99 | 91.84 | 78.05 | 54.95 | 74.95 | 88.23 | 82.69 | 52.13 | 74.35 |
| | 30% | 90.50 | 75.62 | 48.11 | 71.41 | 87.03 | 80.39 | 47.42 | 71.62 | 91.89 | 78.71 | 56.89 | 75.83 | 88.30 | 82.41 | 51.66 | 74.12 |
| | 40% | 90.42 | 76.03 | 47.72 | 71.39 | 87.54 | 81.64 | 48.28 | 72.49 | 91.99 | 79.00 | 56.78 | 75.93 | 88.54 | 81.88 | 51.24 | 73.88 |
| **B.HighC2Low** | 10% | 91.12 | 75.14 | 47.53 | 71.26 | 87.38 | 80.38 | 47.47 | 71.74 | 91.83 | 76.99 | 53.89 | 74.24 | 88.39 | 81.98 | 51.13 | 73.84 |
| | 20% | **91.19** | 75.84 | 48.45 | 71.83 | 87.43 | 82.18 | 48.76 | 72.79 | 92.18 | 78.36 | 56.11 | 75.55 | 88.56 | 82.81 | 51.90 | 74.42 |
| | 30% | 90.72 | 75.86 | 48.15 | 71.58 | 87.45 | 81.71 | 48.03 | 72.40 | 91.99 | 79.20 | 56.89 | 76.03 | 88.51 | 83.51 | 52.13 | 74.72 |
| | 40% | 90.55 | 77.11 | 48.92 | 72.19 | 87.11 | **82.32** | **49.97** | **73.13** | 91.76 | 79.37 | 57.17 | 76.10 | 88.42 | 83.55 | 52.72 | 74.90 |
| **C.Redundant2Low** | 10% | 90.52 | 75.05 | 47.31 | 70.96 | 87.37 | 81.30 | 47.72 | 72.13 | 91.78 | 77.87 | 54.62 | 74.76 | 88.94 | 83.12 | 50.14 | 74.07 |
| | 20% | 90.71 | 76.35 | 48.02 | 71.69 | 87.37 | 81.28 | 48.59 | 72.41 | 91.82 | 79.23 | 57.11 | 76.05 | 88.66 | 84.07 | 51.89 | 74.87 |
| | 30% | 90.45 | 76.70 | 49.26 | 72.14 | 87.27 | 81.62 | 47.59 | 72.16 | 91.80 | 79.03 | 56.70 | 75.84 | 88.66 | 82.52 | 51.88 | 74.35 |
| | 40% | 90.42 | 77.34 | 48.07 | 71.94 | 86.97 | 82.17 | 48.41 | 72.52 | 91.54 | 79.52 | 58.06 | 76.38 | 88.67 | 84.43 | 53.49 | 75.53 |
| *Dataset Transferability Study (Discussed in Section 4.5)* | | | | | | | | | | | | | | | | | |
| **MINER + HighC2Low 40%** | | 90.71 | 79.85 | 53.42 | 74.66 | 87.01 | 80.75 | 49.87 | 72.55 | 91.70 | 80.33 | 61.16 | 77.73 | 89.37 | 85.16 | 54.54 | 76.36 |
| **BERT.HighC2Low 40%** | | – | – | – | – | – | – | – | – | 91.74 | 79.18 | 55.15 | 75.36 | 89.11 | 83.37 | 52.23 | 74.90 |

Table 10: Detailed performance of different approaches to reduce the $\mathcal{V}$-information of entity in the datasets. Each reported result is averaged by 3 repeated experiments.

| Methods | | Base Model: BERT-base-cased | | | | | | | | Base Model: RoBERTa-large | | | | | | | |
|---|---|---|---|---|---|---|---|---|---|---|---|---|---|---|---|---|---|
| | | CoNLL2003 | | | | ACE2005 | | | | CoNLL2003 | | | | ACE2005 | | | |
| | | Test | OOV | Cate. | Avg. | Test | OOV | Cate. | Avg. | Test | OOV | Cate. | Avg. | Test | OOV | Cate. | Avg. |
| **Base** | | **91.18** | 70.89 | 44.48 | 68.85 | 87.48 | 79.80 | 44.84 | 70.71 | 92.10 | 72.54 | 50.01 | 71.55 | 88.84 | 80.84 | 46.13 | 71.94 |
| **DataAug** | | 90.52 | 73.45 | 46.39 | 70.12 | 87.16 | 80.99 | 44.58 | 70.91 | 91.71 | 76.93 | 52.18 | 73.60 | 88.96 | 81.95 | 46.03 | 72.31 |
| **LPFT** | | 91.14 | 73.89 | 48.32 | 71.12 | **87.76** | 80.28 | 44.37 | 70.80 | 92.12 | 75.67 | 48.53 | 72.11 | 88.85 | 82.10 | 46.23 | 72.39 |
| **MINER** | | 90.98 | 76.89 | 49.97 | 72.61 | 87.41 | 79.23 | 45.45 | 70.70 | 91.88 | 79.33 | 56.27 | 75.83 | 88.83 | 82.69 | 46.75 | 72.76 |
| **I.Random2High** | **10%** | 90.74 | 72.41 | 45.59 | 69.58 | 87.17 | 80.19 | 44.69 | 70.68 | 91.60 | 75.78 | 53.22 | 73.53 | 88.97 | 81.51 | 45.96 | 72.15 |
| | **20%** | 90.55 | 73.45 | 47.33 | 70.44 | 87.25 | 80.67 | 45.11 | 71.01 | 91.73 | 77.66 | 54.73 | 74.71 | 88.72 | 83.43 | 46.74 | 72.96 |
| | **30%** | 90.95 | 75.31 | 48.27 | 71.51 | 86.74 | 80.70 | 45.03 | 70.83 | 91.88 | 77.66 | 54.52 | 74.69 | 88.67 | 84.66 | 47.70 | 73.67 |
| | **40%** | 90.43 | 76.69 | 50.20 | 72.44 | 86.87 | 81.21 | 45.26 | 71.12 | 91.67 | 77.86 | 54.85 | 74.79 | 88.53 | 84.08 | 48.81 | 73.81 |
| **II.Low2High** | **10%** | 90.74 | 75.02 | 47.96 | 71.24 | 87.10 | 80.09 | 46.38 | 71.19 | 91.45 | 77.74 | 56.71 | 75.30 | 89.08 | 82.15 | 50.13 | 73.79 |
| | **20%** | 90.48 | 76.51 | 49.90 | 72.30 | 87.05 | 80.79 | 47.81 | 71.88 | 91.30 | 79.33 | 59.48 | 76.70 | 88.87 | 84.13 | 51.93 | 74.97 |
| | **30%** | 90.42 | 76.87 | 52.15 | 73.15 | 86.99 | 81.38 | 46.78 | 71.71 | 90.76 | 79.12 | 60.52 | 76.80 | 88.86 | 83.60 | 50.57 | 74.34 |
| | **40%** | 89.72 | 78.41 | 55.70 | 74.61 | 86.48 | 82.01 | 47.78 | 72.09 | 90.92 | 79.58 | 61.93 | 77.47 | 88.67 | 84.40 | 52.20 | 75.09 |
| *Dataset Transferability Study* (Discussed in Section 4.5) | | | | | | | | | | | | | | | | | |
| **MINER + Low2High 40%** | | 90.18 | 79.69 | 56.03 | 75.30 | 87.13 | 80.16 | 48.72 | 72.00 | 91.72 | 80.05 | 61.03 | 77.60 | 89.33 | 84.67 | 50.07 | 74.69 |
| **BERT.Low2High 40%** | | – | – | – | – | – | – | – | – | 91.05 | 79.63 | 59.79 | 76.82 | 88.79 | 83.28 | 51.88 | 74.65 |

Table 11: Detailed performance of different approaches to enhance the $\mathcal{V}$-information of context in the datasets. Each reported result is averaged by 3 repeated experiments.