# OpenReview forum: "Towards Building More Robust NER datasets: An Empirical Study on NER Dataset Bias from a Dataset Difficulty View"
_EMNLP/2023/Conference — EMNLP 2023 Main_

### Official Review · Reviewer_zR24 · 2023-07-21

**Typos Grammar Style And Presentation Improvements:** 1. I found many typos and grammatical…
**Soundness:** 3

**Excitement:**

4: Strong: This paper deepens the understanding of some phenomenon or lowers the barriers to an existing research direction.

**Paper Topic And Main Contributions:**

This paper provides thorough analyses of dataset bias in named entity recognition (NER). It introduces V-information for measuring the difficulty of individual examples and entire datasets and context-entity information margin (CEIM) for assessing the difference in difficulty between entity- and context-level instances. Based on the analyses, the paper found that lexical information of entities is easier to learn compared to contextual information, which makes models learn shortcuts from entity strings. The paper argues that the difference in learning difficulty between entity and context information should be reduced to de-bias the model. The proposed ideas include increasing the portion of difficult entities or increasing the portion of easy contexts in the training data, which improved the performance of baseline models in several robustness tests on NER benchmark datasets.

**Reasons To Accept:**

1. The analysis methods, V-information and CEIM, are first used in the field of NER by this paper and are technically sound.

2. The paper is firmly grounded in, and well-supported by relevant studies.

3. The results are interesting. In Tables 3 and 4, although the proposed methods do not improve the performance on in-domain test sets, they are effective on out-of-domain or out-of-vocabulary test sets, even when applied to the MINER model.

**Reasons To Reject:**

1. The presentation quality should be improved. Please see the comments in the section "Typos Grammar Style And Presentation Improvements."


**Reproducibility:**

3: Could reproduce the results with some difficulty. The settings of parameters are underspecified or subjectively determined; the training/evaluation data are not widely available.

**Reviewer Confidence:**

4: Quite sure. I tried to check the important points carefully. It's unlikely, though conceivable, that I missed something that should affect my ratings.

---

> ### Author Rebuttal · Authors · 2023-08-29
>
> We sincerely appreciate your thorough reading and constructive feedback!
>
> **Regarding the presentation quality:**
>
> We apologize for the presentation quality and all the mistakes we made. Thank you very much for your careful review and pointing out all these errors. We will thoroughly review and revise the paper based on your review. Also, the suggestions to include Appendix A.1 and Table 5 in the main page, and to increase the table/figure sizes are very helpful. We will take your advice once we have enough pages. Thanks again!

---

### Official Review · Reviewer_keJx · 2023-08-04

**Soundness:** 4
**Typos Grammar Style And Presentation Improvements:** 065

**Excitement:**

4: Strong: This paper deepens the understanding of some phenomenon or lowers the barriers to an existing research direction.

**Missing References:**



Though not in the same area, the paper "Learning from Context or Names? An Empirical Study on Neural Relation Extraction" already studied a similar "entity or context" problem in relation extraction.


**Paper Topic And Main Contributions:**

This paper studies the named entity recognition (NER) robustness problem from a dataset difficulty perspective. Previous work shows that models tend to have short-cut learning via entity names and there is literature constructing adversarial datasets detecting such short-cut behavior. The authors of this paper study this problem via V-information (or dataset difficulty) instead.

The idea is straightforward: the authors measure how difficult it is to make predictions purely based on entities and purely based on context. And the results on several commonly-used NER datasets show that using entity names alone is even easier, which suggests shortcuts. With this observation, the authors propose several data-augmentation methods by replacing easy-entity / hard-context (to discourage the models to do short-cut learning) and the proposed methods improve models' performance on adversarial test sets.


**Questions For The Authors:**


(1) The used datasets are quite old. There are newer datasets like FewNerd which has more training examples and NER types. Using datasets with more types might be better for this paper's purpose.

(2) In Table 4, it seems that the green line performance (even in the OOV case) can't match using 10,000 examples. I wonder how the proposed data augmentation method would perform when using more instances. Would there be diminishing returns?



**Reasons To Accept:**

(1) The authors study the robust NER problem from a new dataset difficulty perspective.

(2) The observation is drawn from comprehensive experiments (multiple NER datasets, detailed analysis, intuitive experiment design).

(3) Based on the observation that easy-entity names caused the shortcut, the authors design data-augmentation methods to alleviate the problems. The proposed methods successfully improved the model's performance on robustness test sets.


**Reasons To Reject:**

(1) Though the V-information angle is new, the idea that "entity names cause shortcuts" has been heavily explored. The contribution is mainly how to present the shortcut (via this new perspective).


**Reproducibility:**

5: Could easily reproduce the results.

**Reviewer Confidence:**

4: Quite sure. I tried to check the important points carefully. It's unlikely, though conceivable, that I missed something that should affect my ratings.

---

> ### Author Rebuttal · Authors · 2023-08-29
>
> We sincerely appreciate your thorough reading and constructive feedback!
>
> #### **Response to Question (1):**
>
> Thank you for your valuable advice about the newer datasets such as Few-NERD. To better illustrate our work, we further conduct several experiments on the Few-NERD dataset. The results are rather interesting, and we are glad to discuss our findings with you:
>
> The Few-NERD has 66 fine-grained categories, which is more fine-grained than any of the existing NER datasets. By measuring the V-information on these fine-grained classes, we find that:
>
> 1. Generally, the fine-grained dataset shows less dataset bias than the coarse-grained dataset: in most of the fine-grained classes, the V-information of the context is comparable and even larger than the V-information of the entity. This might be because:
>     1. Compared with categorizing coarse-grained classes (such as “person”, “location”, “organization”), categorizing different entities between fine-grained classes (such as “person-scholar”, “person-actor”, “person-artist”) is inherently harder, where the contexts include more clues for categorization.
>     2. The Few-NERD dataset is much larger than most existing datasets (about 130k training data) and contains more classes, and the entity patterns of each class are much more complicated to learn. For each class, it is harder for the model to predict the entity class through shortcut learning certain entity patterns.
> 2. The analysis method we proposed allows us to delve into the distribution of different fine-grained classes, and we find that the V-information distribution is quite diverse across different classes:
>     1. In most classes, the V-information of entity and context are comparable. As for the CEIM distribution, the near-zero CEIM data still cover the largest proportion of most classes (the high-CEIM and low-CEIM proportion is relatively balanced).
>     2. In some classes, the V-information of entity is rather high, such as the “location-park” (V-info entity/context=6.04/4.23), “building-hospital” (V-info entity/context=6.84/5.93), etc. This might be because the entities in these classes generally show simple patterns. For example, most entities in “location-park” are ended with “park”.
>     3. Meanwhile, there exist some classes that heavily rely on the context information for prediction, such as the “person-athlete” (V-info entity/context=2.29/4.30), “person-actor” (V-info entity/context=2.79/4.80) class, etc. These classes, such as the “person-athlete” class, mostly consist of a normal person name, and it is quite hard for the model to deduce the correct class without the context. For instance, in the sentence “Honky then lost his championship at Summerslam”, one cannot tell “honky” is an athlete without the context.
> 3. Based on findings 1,2, we conclude that Few-NERD is a much more robust dataset than most of the existing coarse-grained datasets. Nevertheless, there still exists severe bias problems in some classes (such as the biased “location-park”, “building-hospital” classes). Also, most classes that “have comparable V-information of entity and context” might also have potential robustness problems according to the CEIM distribution. **Based on the detailed analysis, we believe that our dataset reconstruction methods can be further applied to improve the robustness of Few-NERD in a class-wise manner.** Unfortunately, we didn’t conduct further robustness tests on the Few-NERD dataset as in Table 2 and Section 4 because **we didn’t have the robustness test sets as on CoNLL03 and ACE2005 (the robustness test sets we use now are constructed by (Wang et al., 2021)).** We agree with you that it will be more convincing to conduct the experiments on Few-NERD, and we also consider it will be interesting to delve into the potential robustness of Few-NERD. In future work, we might try to construct our own robustness test sets on Few-NERD and conduct more robustness studies. Thanks again for your advice!
> 4. Another interesting little finding: We think Point 2 (3) also reveals that the model robustness (corresponding to learning more context) might also be important when we need to generalize the NER model to recognize more fine-grained classes that highly rely on context information. To verify this, we further conducted two simple transfer experiments:
>     1. Model A: first trained the NER model on the original CoNLL03 dataset, and then adapted the model to the Few-NERD dataset via few-shot training;
>     2. Model B: first trained the NER model on our reconstructed CoNLL03 datasets, and then adapted the model to the Few-NERD dataset via few-shot training.
>
>     We find that model B that trained on the reconstructed robust dataset can achieve better few-shot results on the context-relied classes (for example, the average 5-shot f1-score of “person-x” classes is 20.01 (model A) and 24.23 (model B)) while maintaining similar results with model A on other classes (for example, the average 5-shot f1-score of “organization-x” classes is 40.07 (model A) and 40.18 (model B)). Although this experiment is rather simple, it indicates the importance of building more robust datasets for model generalization.
>
>
> #### **Response to Question (2):**
>
> In Figure 4 (do you mean Figure 4 rather than Table 4?), we only conducted the data proportion altering studies on 1000 reconstructed data (this is because there are very few low-CEIM data in the datasets and it’s hard to achieve 90% low-CEIM proportion on the full datasets). Therefore, it is straightforward that the green line performance cannot match the performance of 10000 data. We compared the green line with more data here only to reveal the potential of the 1000 reconstructed data. We also admit the limitation of this proportion-altering method since it is hard to apply to larger data.
>
> Then, in order to further validate the data reconstruction methods when using more instances, in Section 4.3, 4.4, we designed two data-size-agnostic data reconstruction methods, and verified their effectiveness on full datasets. **All experiments in Table 3 and Table 4 are conducted on the full CoNLL03 and ACE 2005 datasets (15k and 9.7k training data, respectively)**. As shown in the tables, the proposed dataset reconstruction methods are still quite helpful when using more instances. Moreover, as discussed in both sections, these two methods actually also result in the increase of low-CEIM data proportion, which is similar to what we directly did in Figure 4. Therefore, these results indicate that enlarging the proportion of low-CEIM data can also be helpful when using more instances.
>
> #### **Response to “Reasons To Reject”:**
>
> We agree with you that the idea that "entity names cause shortcuts" has been heavily explored, and our work is mainly on how to present the shortcut via the new V-information perspective. Despite this, we would like to further discuss our contribution:
>
> 1. Previous work actually focus more on how the “entity distribution” affects NER robustness. While in this work, we are the first to comprehensively consider the joint distribution of entity and context on this problem. For one thing, the V-information-based measures we introduced allow us to systematically examine and “compare” the entity and context difficulty distribution, thus explaining the dataset bias in a more comprehensive manner. For another, we are the first to consider “improving robustness via altering context distribution” and validate the effectiveness of altering context distribution on model robustness (Section 4.4, Table 4). These results are rather intuitive based on our preliminary analysis in Section 2, yet are very interesting and provide new insights on how to build better NER datasets from the context view.
> 2. Previous work on NER robustness mainly focus on discovering or verifying the robustness problem via experimenting, or proposing methods to improve model robustness. In this work, we offer a more systematic view for examining and analyzing the NER dataset bias and robustness problem, and verify that our new view can induce valid new methods for improving NER robustness (such as the context-based method). More importantly, we hope all these new views, quantified analysis, and exploratory experiments can provide useful references for “how to build more robust NER datasets”. For one thing, people can use the measures we used to assess existing NER datasets regarding how robust these datasets are. For another, people can refer to our measures and methods to improve existing datasets they have, and the improved datasets are verified to be transferable (Table 3,4). Furthermore, when building new NER datasets, our work might also be a good reference for raw data selection (e.g., to select which instances should be annotated in order to build more robust datasets. So far we haven’t explored methods toward this goal, yet we consider it will be interesting future work.) We believe our work is meaningful and hope it can provide distinctive insight into building more robust NER datasets.
>
> #### **Response to the Missing References:**
>
> Thank you very much for reminding us of this missing reference! Actually, this interesting work, although not in the same area, also motivates us in the experiment designs. We will add it to the Related Work section in the next version.
>
> #### **Regarding typos errors:**
>
> Thanks for pointing out these problems! We will carefully examine and revise our paper based on your advice in the next version.

---

### Official Review · Reviewer_kCPH · 2023-08-05

**Typos Grammar Style And Presentation Improvements:** See reasons to reject.
**Soundness:** 4

**Excitement:**

3: Ambivalent: It has merits (e.g., it reports state-of-the-art results, the idea is nice), but there are key weaknesses (e.g., it describes incremental work), and it can significantly benefit from another round of revision. However, I won't object to accepting it if my co-reviewers champion it.

**Missing References:**

See reasons to reject.

**Paper Topic And Main Contributions:**

The paper conducts an interesting study on the model robustness problem of NER. The paper quantifies the entity-context difficulty distribution in the existing dataset using V-information and CEIM and finds that the entity is much easier for the pre-trained model to learn than the context. Meanwhile, the paper explains the relationship between the entity-context difficulty and model robustness and finds the low-CEIM instances, i.e., instances with contexts easier than entities, contribute most to the model robustness. Based on the findings, the paper suggests three potential ways to de-bias the NER datasets by altering the entity-context distribution. The experimental results show that the de-biased datasets can benefit different models.

**Questions For The Authors:**

See reasons to reject.

**Reasons To Accept:**

1. The paper presents a comprehensive study of NER dataset bias and model robustness from a dataset difficulty view, which may provide some new insights for NER research.
2. The paper proposes three potential ways to de-bias the NER datasets by introducing more instances with low-CEIM. The experimental results demonstrate the effectiveness of the proposed methods.

**Reasons To Reject:**

1. Motivation: The paper mentions there are many previous works that have delved into the robustness problem of NER, including the studies investigating how the contexts and entities affect the performance. So what are the specific differences between this paper and previous work? Has the conclusion of the paper, i.e., "the entity is much easier for the pre-trained model to learn than the context", been discovered in previous work？
2. Presentation: (1) There are many terms in the text that are not explained very clearly. For example, Does the term "data bias" mean instances with high-CEIM? If yes, in real-world applications, there are many instances, e.g., "Washington", where we can directly determine the type of entity by its name. The term "entity-context distribution" is also not clear. (2)  In Figure 1, the paper points out two typical failures. However, the paper does not seem to discuss the second type of error ("ambiguous entity"). (3) There are many typos or mistakes in the paper. For example, there lack some citations in line 68. In sec. 3.3 line 285, Is the ''Table 2'' refers to "Table 1"?

**Reproducibility:**

4: Could mostly reproduce the results, but there may be some variation because of sample variance or minor variations in their interpretation of the protocol or method.

**Reviewer Confidence:**

4: Quite sure. I tried to check the important points carefully. It's unlikely, though conceivable, that I missed something that should affect my ratings.

---

> ### Author Rebuttal · Authors · 2023-08-29
>
> We sincerely appreciate your thorough reading and constructive feedback!
>
>
> #### **Regarding the motivation:**
>
> Sorry for claiming our motivation and main contribution less clearly. It is true that “the entity is much easier for the pre-trained model to learn than the context” has also been discovered and discussed in previous work. However, we want to clarify that this is not the “conclusion” of our work. As shown in line 076-082, we regard this point more as the motivation or verified support for our work. In fact, we start with previous findings on “the entity is easier to learn”, trying to provide a more systematic and more comprehensive investigation on this point. We want to reclaim the differences between this work and previous work as follows:
>
> 1. While most previous work only verified or discussed this point experimentally (such as testing models on challenging test sets [1], conducting randomization experiments [2], etc. ), we are the first to systematically quantify this point from a theoretically-verified perspective (i.e., the V-information), providing a new angle of explanation for the NER robustness and the dataset bias.
> 2. While previous work mainly focus on the impact of entity distribution on the model robustness, we consider the context distribution is also significant regarding the dataset bias, and the impact of entity cannot be isolated without considering the context. With the dataset-difficulty view, we are able to quantify and “compare” the difficulty of entity and context, and conduct a more comprehensive investigation into the joint distribution of entity and context, as well as how different joint distribution induces dataset bias. **We are the first to comprehensively consider the joint distribution of context and entity towards NER robustness.** (In Table 4, we verified that context distribution is indeed important for model robustness. We are the first to verify this and successfully improve the NER robustness by altering context distribution.)
> 3. Besides how we provide new (and more comprehensive) explanation to this point, we also verified that our explanation can serve as a successful tool for constructing more robust datasets. While some previous works also conduct data augmentation for improving model robustness, their practices are relatively heuristic and empirical (such as switching entities to similar ones) and lack effective measures and targets. **We provide the dataset reconstruction with quantified measures (the PVI and the V-information) and verified targets (e.g., to enhance the V-information of context)**, and we believe our work is helpful for future work to evaluate the dataset robustness, refine existing datasets, and even build new robust datasets.
>
>
>
> #### **Regarding the Presentation:**
>
> Sorry for presenting the paper less clearly and causing confusion. We would like to explain the confusing points as follows:
>
> 1. Regarding “data bias”, “high-CEIM” and “entity-context distribution”:
>     1. The “data bias” (or “dataset bias”) refers to the intrinsic bias in the dataset that might induce shortcut learning. Specifically, in NER dataset, if the entity itself is adequate for recognizing the correct class, and the entity itself is much easier to learn and remember than the context, the model will not learn to deduce the answer using context information. This causes a shortcut learning problem of NER models, which will harm model robustness and generalization on other domains or OOD data. For example, if in the training data, “Washington” is mostly labeled as a Person name, the model will learn to directly regard “Washington” as a Person name without referring to the context. Then, if the model needs to recognize entities in the sentence “He lives in *Washington*”, it will incorrectly recognize “Washington” as Person, regardless that the context indicates “Washington” is a location.
>     2. By calculating and experimenting with the CEIM, we recognize two kinds of CEIM data that might induce dataset bias, i.e., the high-CEIM and near-zero-CEIM data. We regard the high proportion of high-CEIM and near-zero-CEIM data as a cause of NER dataset bias, and verified that deducing the proportion of these data helps alleviate dataset bias (corresponding to better model robustness).
>     3. The “entity-context distribution” refers to the “difficulty” distribution of the entity, the context, and the joint difficulty distribution of entity and context. (For instance, how many “easy” or “difficult” entities/contexts are there in the data? In each single sentence, how difficult is the entity and the context, and which one is easier to learn? In the whole dataset, how many instances contain an entity easier/more difficult than the context? As for the whole dataset, is the entity information easier for the model to learn than the context information?) By systematically examining the entity-context distribution towards difficulty, we are able to explain the NER dataset bias from a more comprehensive view, as discussed before.
> 2. In Figure 1, we point out two typical failures, i.e., the OOV (out-of-vocabulary) failure and the ambiguous entity failure. **Both these two types of errors are discussed in the paper.** In line 237-243, we declared that the adopted two robustness test sets exactly correspond to the two failures, respectively. We also show some cases of the two robustness test sets in Appendix A.2. In Table 9, we can see that the OOV test set is designed to evaluate the OOV failure, which introduces many OOV entities. Similarly, the Cross-category (”Cate.”) test set introduces many ambiguous entities to challenge the model. All our robustness experiments are tested on these two test sets to show the model robustness towards both OOV and ambiguous entity failures.
> 3. Thank you for pointing out the typos and mistakes in our paper! We will thoroughly review and correct all these typos/mistakes in the next version of this paper.
>
>
>
> We apologize again for all our misleading and confusing writing and thank you very much for pointing out all these problems. These feedbacks are quite helpful for us to improve our paper!
>
> [1] Ghaddar, A., Langlais, P., Rashid, A., & Rezagholizadeh, M. (2021). Context-aware adversarial training for name regularity bias in named entity recognition. *Transactions of the Association for Computational Linguistics*, *9*, 586-604.
>
> [2] Lin, Hongyu, et al. "A Rigorous Study on Named Entity Recognition: Can Fine-tuning Pretrained Model Lead to the Promised Land?." *Proceedings of the 2020 Conference on Empirical Methods in Natural Language Processing (EMNLP)*. 2020.

---

### Meta-Review · Area_Chair_i9Qm · 2023-09-19

**Recommendation:** 3

**Metareview:**

The paper conducts an interesting study on the model robustness problem of NER. It introduces V-information for measuring the difficulty of individual examples and entire datasets and context-entity information margin (CEIM) for assessing the difference in difficulty between entity- and context-level instances.
the paper found that lexical information of entities is easier to learn compared to contextual information, which makes models learn shortcuts from entity strings. The paper argues that the difference in learning difficulty between entity and context information should be reduced to de-bias the model.  In general, this paper represents a well-executed and systematic exploration. Its main limitation lies in the lack of novelty in the conclusions it draws and the proposed solutions, as they align closely with widely accepted findings within the field.

---

### Decision · Program_Chairs · 2023-10-07

**Decision:**

Accept-Main

**Comment:**

The paper conducts an interesting study on the model robustness problem of NER. It introduces V-information for measuring the difficulty of individual examples and entire datasets and context-entity information margin (CEIM) for assessing the difference in difficulty between entity- and context-level instances.
the paper found that lexical information of entities is easier to learn compared to contextual information, which makes models learn shortcuts from entity strings. The paper argues that the difference in learning difficulty between entity and context information should be reduced to de-bias the model.  In general, this paper represents a well-executed and systematic exploration. Its main limitation lies in the lack of novelty in the conclusions it draws and the proposed solutions, as they align closely with widely accepted findings within the field.